# Adaptive Polynomials for the Vibration Analysis of an L-Type Beam Structure with a Free End

Duck Young Yoon [1] and Jeong Hee Park [2,]*

1 Departament of Naval Architecture and Ocean Engineering, Chosun University, Gwangju 61452, Korea; dyyun@chosun.ac.kr
2 Structural Design Departament, Hyundai Samho Heavy Industries, Jeollanam-do 58462, Korea
* Correspondence: _for_dream@hanmail.net

**Abstract:** Vibration analysis using the component mode method has been less popular than before, since computers are powerful enough to solve complicated structures by a single large finite model. However, many structural engineers designing local structures on a ship still need simple tools to check anticipated vibration problems during their design work. Since most of local structures on a ship are simple enough to consist of several substructures, the component mode method could be of use as long as good, natural mode functions can be provided so that reasonable natural frequencies can be yielded. In this study, since mode polynomials based on static deflection of cantilever beams fail to work to cover the various configurations of L-type beams with a free end, two alternatives are suggested. One is based on more flexible mode functions—we call them adaptive polynomials. The other is a purely mathematical approach, which makes realistic mode functions unnecessary. Suggested alternatives yield very good numerical results.

**Keywords:** L-type beam structure; adaptive polynomials; pure mathematical functions



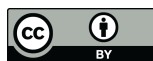

## 1. Introduction

Two component mode methods are well-known: component mode synthesis, suggested by Hurty [1,2] and Craig [3,4], and the branch mode method, suggested by Hunn [5] and Gladwell [6].

The L-type beam structure with a free end studied here has been used for the explanation of the branch mode method, where normal modes of cantilevers, together with rigid mode, are suggested for generating branch modes [7].

Bhat suggested higher mode functions using orthogonal polynomials, and applied these mode functions for the free vibration analysis of a single plate with different boundary conditions [8].

Bourquin [9], Hou [10], Hintz [11], and Benfield [12] have studied the constraints at the junction of the connected structures.

Recent studies have focused on nonlinear mechanics. Pagani et al. explained that the natural frequency and mode shape can be changed significantly when the metal structure is subjected to large displacement and rotation under geometrical nonlinear conditions [13]. Carrera et al. developed the Lagrange formula, including cross-sectional deformation, in order to implement the vibration mode of the composite beam structure in the nonlinear region [14].

Furthermore, Carrera et al. developed a theory that can be solved by converting a three-dimensional model for large deformation of a structure into one dimension, which was developed and applied to the calculation [15].

Pagani and Carrera [16] introduced the unified formulation for geometric, nonlinear analysis of metal structures, and explained the formula for handling large displacements and rotations. In order to solve the geometric nonlinear problem of the plate, Pagani et al. [17]

explained various nonlinear theories, and how these theories affect the nonlinear static behavior of thin-walled structures in the large displacement and rotation.

Alessandro et al. [18] introduced application of the fundamental model reduction techniques used in structural dynamics to flexible, multibody systems.

Park [19] suggested mode functions for L-type beam structures with fixed ends, where constraints at a junction are described using fixed and simple supported boundary conditions. An application of this mode function for plate structure has also been found [20].

As mentioned, if suitable mode functions are available, the component mode method can be a powerful tool for the free vibration analysis of simple local structures.

The purpose of this study is to provide powerful mode functions for the free vibration analysis of L-type beam structures, as shown in Figure 1, which can work for various length ratios ($0 \leq L_B/L_A \leq \infty$). $L_A$ and $L_B$ are the lengths of the connected L-type beam in Figure 1.

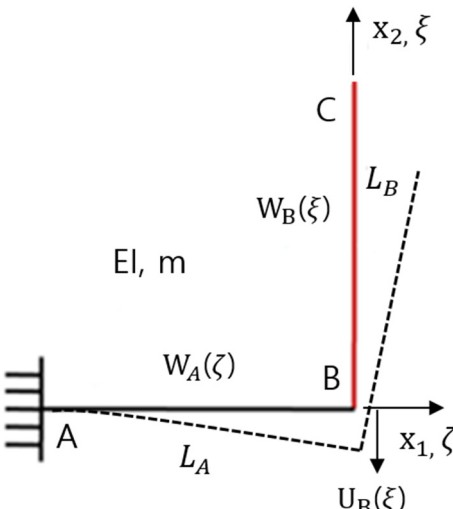

**Figure 1.** Simplified model of one end of a supported structure.

Fundamental mode function, based on a fourth-order polynomial that satisfies four boundary conditions of a cantilever beam, failed to work for free vibration analysis of an L-type beam structure with a free end, although it describes deflections of a cantilever beam reasonably well. It is a reasonable guess that any mode functions that can describe well the deflections of substructures, which are cantilever beams, may not be able to describe deflections of the L-type beam structure with a free end.

New fundamental mode functions, using a second-order polynomial together with higher orthogonal mode functions, are suggested. These new mode functions have been found to be suitable for the free vibration analysis of L-type beam structures for various length ratios ($0 \leq L_B/L_A \leq \infty$). This good performance is because of the fact that new mode functions based on lower-order polynomials are flexible enough to describe various shapes of deflections of varying length ratios. In this sense, these new polynomials can be named as adaptive polynomials.

In addition, a purely mathematical approach is suggested, where no efforts to describe meaningful mode functions are necessary. Instead, pure mathematical polynomials that only satisfy geometrical boundary conditions at a free end are used.

## 2. Problem Description and Mathematical Model

An L-type beam structure with a free end is shown in Figure 1:

Where $m_A = m_B = m$, $EI_A = EI_B = EI$ are assumed to be same for notational simplicity; *m* is mass per unit length of beam, and *E* and *I* are the Young's modulus and moment of inertia, respectively.

$x_1$ and $x_2$ are coordinates of substructrues of L-type structures in Figure 1.

In addition, $x_1$ and $x_2$ are non-dimensionalized, such that $\zeta = \frac{x_1}{L_A}$, $\xi = \frac{x_2}{L_B}$.

$W_A(\zeta)$ and $W_B(\xi)$ are lateral deflections of the horizontal and vertical beam, respectively:

$$W_A(\zeta, t) = \sum_{i=1}^{m} \phi_i(\zeta)p_i(t) \tag{1}$$

$$W_B(\xi, t) = \sum_{j=1}^{n} \psi_j(\xi)q_j(t) \tag{2}$$

$$U_B(\xi, t) = r_1(t) \tag{3}$$

where $p_i(t), q_j(t)$, and $r_1(t)$ are the generalized coordinates, and $\phi_i(\zeta)$ and $\psi_j(\xi)$ are corresponding mode functions to describe lateral deflections of beams A and B. $U_B(t)$ is vertical displacement of beam B.

The method proposed by Bhat to generate higher orthogonal polynomials is as follows:

$$\phi_2(\zeta) = (\zeta - B_1)\phi_1(\zeta) \tag{4}$$

$$\phi_k(\zeta) = (\zeta - B_k)\phi_{k-1}(\zeta) - C_k\phi_{k-2}(\zeta) \tag{5}$$

$$B_k = \int_0^1 \zeta \cdot \phi_{k-1}^2(\zeta)d\zeta \Big/ \int_0^1 \phi_{k-1}^2(\zeta)d\zeta \tag{6}$$

$$C_k = \int_0^1 \zeta \cdot \phi_{k-1}(\zeta)\phi_{k-2}(\zeta)d\zeta \Big/ \int_0^1 \phi_{k-2}^2(\zeta)d\zeta \tag{7}$$

It can be shown that the polynomial $\phi_k(\zeta)$ satisfies the orthogonality condition:

The coefficients $B_k$ and $C_k$ are implemented using the orthogonal formula of the beam function:

$$\int_0^1 \phi_k(\zeta)\phi_l(\zeta)d\zeta = \left\{ \begin{array}{l} 0 \; if \; k \neq l \\ 1 \; if \; k = l \end{array} \right\} \tag{8}$$

Note that this polynomial $\phi_k(\zeta)$ only satisfies geometrical boundary conditions, although a fundamental polynomial can be chosen to satisfy natural boundary conditions.

Given mode functions, generalized mass is

$$m_{Aij} = mL_A \int_0^1 \phi_i\phi_j d\zeta \tag{9}$$

$$m_{Bij} = mL_B \int_0^1 \psi_i\psi_j d\xi \tag{10}$$

$$m_{Brr} = mL_B \int_0^1 d\xi \tag{11}$$

and generalized stiffness ($k$) is

$$k_{Aij} = \frac{EI}{L_A^3} \int_0^1 \phi''_i\phi''_j d\zeta \tag{12}$$

$$k_{Bij} = \frac{EI}{L_B^3} \int_0^1 \psi''_i\psi''_j d\xi \tag{13}$$

where, $m_{Aij}$, $m_{Bij}$, $m_{Brr}$ and $k_{Aij}$, $k_{Bij}$ are generalized mass and generalized stiffness of substructures of L-type structure.

Applying displacement continuity ($r_1$) at the junction results in the following:

$$r_1 = \sum \phi_i(1) p_i = (\phi_1(1) p_1 + \phi_2(1) p_2 + \cdots + \phi_m(1) p_m) \tag{14}$$

Similarly, applying slope continuity ($q_n$) yields,

$$q_n = \frac{L_B}{L_A} \left( \frac{\phi'_1(1)}{\psi'_n(0)} p_1 + \frac{\phi'_2(1)}{\psi'_n(0)} p_2 + \cdots + \frac{\phi'_m(1)}{\psi'_n(0)} p_m - \frac{\psi'_1(0)}{\psi'_n(0)} q_1 - \frac{\psi'_2(0)}{\psi'_n(0)} q_2 - \cdots - \frac{\psi'_{n-1}(0)}{\psi'_n(0)} q_{n-1} \right) \tag{15}$$

The mass and stiffness matrices ($M_A$ and $K_A$, respectively) using the suggested polynomials are shown in Equations (16) and (17):

$$[M_A] = mL_A \begin{vmatrix} \phi_1\phi_1 & \cdots & \phi_1\phi_m & \phi_1\psi_1 & \cdots & \phi_1\psi_n \\ \vdots & \ddots & \vdots & \vdots & \ddots & \vdots \\ \phi_m\phi_1 & \cdots & \phi_m\phi_m & \phi_m\psi_1 & \cdots & \phi_m\psi_n \\ \psi_1\phi_1 & \cdots & \psi_1\phi_m & \psi_1\psi_1 & \cdots & \psi_1\psi_n \\ \vdots & \ddots & \vdots & \vdots & \ddots & \vdots \\ \psi_n\phi_1 & \cdots & \psi_n\phi_m & \psi_n\psi_1 & \cdots & \psi_n\psi_n \end{vmatrix} \tag{16}$$

$$[K_A] = \frac{8EI}{L_A^3} \begin{vmatrix} \phi''_1\phi''_1 & \cdots & \phi''_1\phi''_m & \phi''_1\psi''_1 & \cdots & \phi''_1\psi''_n \\ \vdots & \ddots & \vdots & \vdots & \ddots & \vdots \\ \phi''_m\phi''_1 & \cdots & \phi''_m\phi''_m & \phi''_m\psi''_1 & \cdots & \phi''_m\psi''_n \\ \psi''_1\phi''_1 & \cdots & \psi''_1\phi''_m & \psi''_1\psi''_1 & \cdots & \psi''_1\psi''_n \\ \vdots & \ddots & \vdots & \vdots & \ddots & \vdots \\ \psi''_n\phi''_1 & \cdots & \psi''_n\phi''_m & \psi''_n\psi''_1 & \cdots & \psi''_n\psi''_n \end{vmatrix} \tag{17}$$

and $M_B$ and $K_B$ can be expressed in a similar manner. Using the displacement and slope continuity in Equations (14) and (15) yields Equation (18):

$$\begin{Bmatrix} p_1 \\ \vdots \\ p_m \\ q_1 \\ \vdots \\ q_{n-1} \\ q_n \\ r_1 \end{Bmatrix} = \begin{vmatrix} \phi_1\phi_1 & \cdots & \phi_1\phi_m & \phi_1\psi_1 & \cdots & \phi_1\psi_n \\ \vdots & \ddots & \vdots & \vdots & \ddots & \vdots \\ \phi_m\phi_1 & \cdots & \phi_m\phi_m & \phi_m\psi_1 & \cdots & \phi_m\psi_n \\ \psi_1\phi_1 & \cdots & \psi_1\phi_m & \psi_1\psi_1 & \cdots & \psi_1\psi_n \\ \vdots & \ddots & \vdots & \vdots & \ddots & \vdots \\ \psi_n\phi_1 & \cdots & \psi_n\phi_m & \psi_n\psi_1 & \cdots & \psi_n\psi_n \\ \alpha\frac{\psi'_1(1)}{\phi'_n(0)} & \cdots & \alpha\frac{\psi'_m(1)}{\phi'_n(0)} & -\alpha\frac{\psi'_1(1)}{\phi'_n(0)} & \cdots & -\alpha\frac{\psi'_{n-1}(1)}{\phi'_n(0)} \\ \phi_1(1) & \cdots & \phi_m(1) & 0 & \cdots & 0 \end{vmatrix} \begin{Bmatrix} p_1 \\ \cdots \\ p_m \\ q_1 \\ \cdots \\ q_{n-1} \end{Bmatrix} \tag{18}$$

where $\alpha$ is the ratio of length for the subcomponents ($\alpha = L_B/L_A$):

## 3. FEM (Finite Element Method) Analysis

For comparison, FEM analysis was performed first. The eam properties used are shown in Table 1, and Figure 2 shows the L-type finite element method (FEM) model and geometric boundary conditions.

**Table 1.** Properties of the finite element method (FEM) model.

| Property | $W_A$ | Cross Section | |
|---|---|---|---|
| Density (kg/m$^3$) | 7850 | | |
| Total Length ($L_A + L_B$) (m) | 10 | | 0.2m |
| Young's modulus (N/mm$^2$) | $2.1 \times 10^5$ | | |
| Moment of inertia (mm$^4$) | $3.33 \times 10^8$ | 0.5m | |

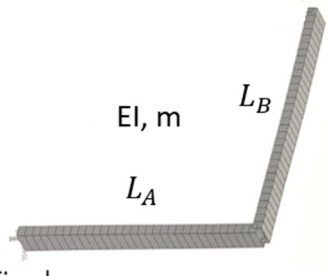

**Figure 2.** L-type beam structure with a free end.

In Figure 2, *m* is mass per unit length of beam, *E* and *I* are Young's modulus and moment of inertia, respectively.

Natural frequencies are shown in Figure 3, Figure 4, Figure 5.

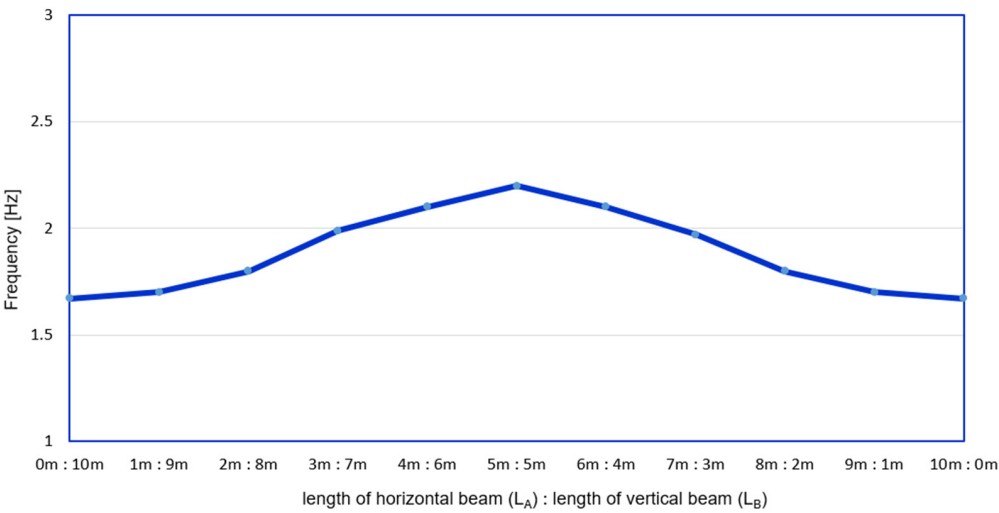

**Figure 3.** Natural frequency of the first mode for an L-type beam structure.

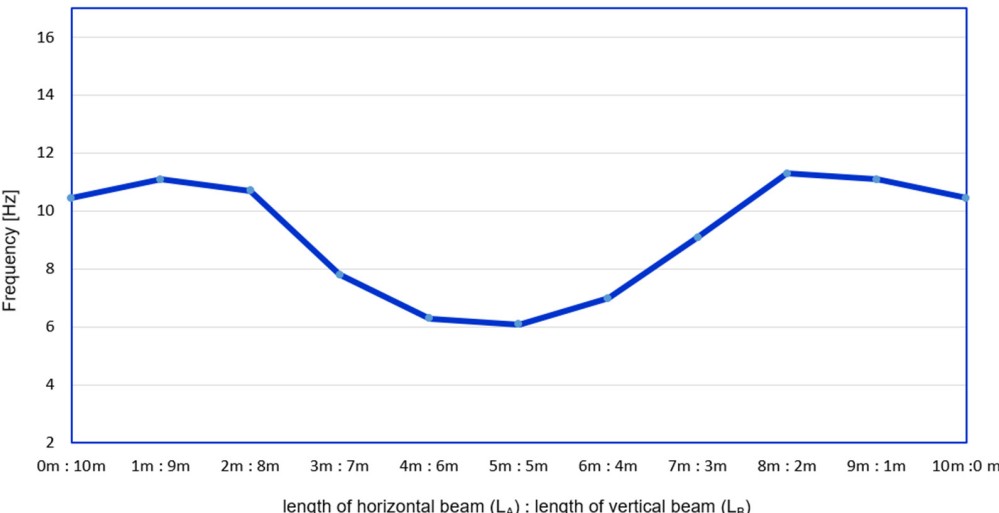

**Figure 4.** Natural frequency of the second mode for an L-type beam structure.

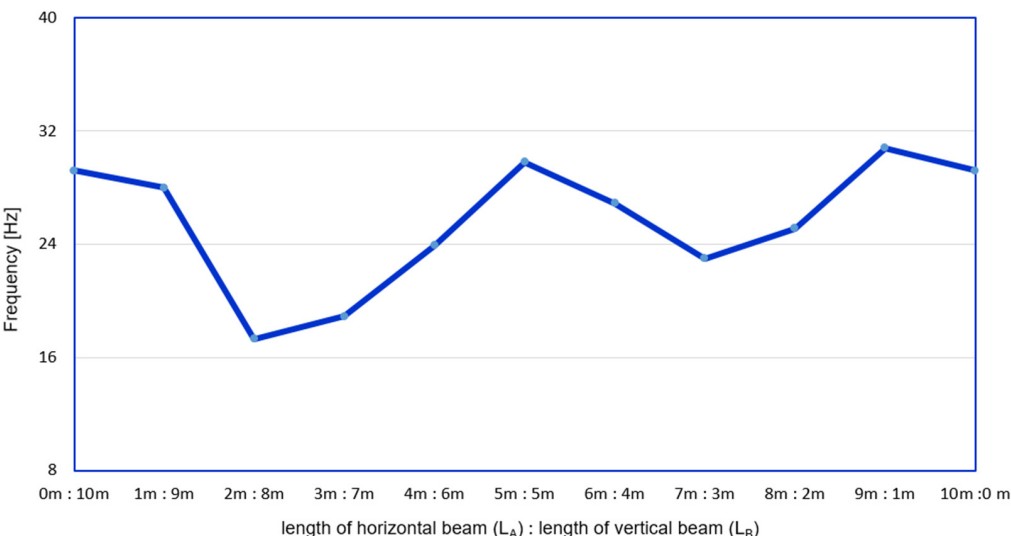

**Figure 5.** Natural frequency of the third mode for an L-type beam structure.

It is worth noting that the natural frequencies for the length ratio $L_B/L_A$ are relatively similar to those for the length ratio $L_A/L_B$ as like Figure 6, although mode shapes are different; this is somewhat interesting. However, it can be understood because this structure becomes a cantilever beam as $L_A$ or $L_B$ approaches zero.

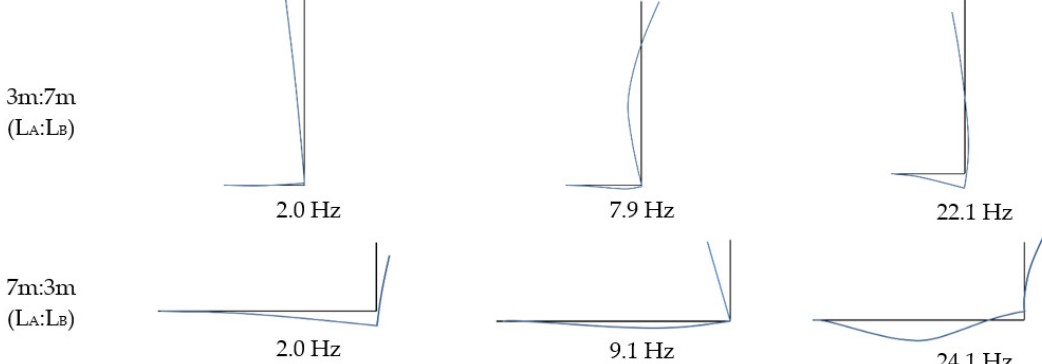

**Figure 6.** Typical mode shape of FEM result: first, second, and third mode.

## 4. Fundamental Mode Function Using Fourth-Order Polynomial and Numerical Results

The fourth-order polynomial for fundamental mode function $\varnothing_1(\zeta)$ can be easily obtained from four boundary conditions of a cantilever beam:

$$w(0) = w'(0) = 0, w''(1) = w'''(1) = 0$$

The lower four polynomials are shown in Table 2.

**Table 2.** The mode function using fourth-order polynomial.

| $i$ | Mode Functions ($\phi_i$) |
|---|---|
| 1 | $\phi_1(\zeta) = \zeta^4 - 4\zeta^3 + 6\zeta^2$ |
| 2 | $\phi_2(\zeta) = \zeta^5 - 4.8022\zeta^4 + 9.2088\zeta^3 - 4.8132\zeta^2$ |
| 3 | $\phi_3(\zeta) = \zeta^6 - 5.4477\zeta^5 + 12.2838\zeta^4 - 10.6580\zeta^3 + 2.6575\zeta^2$ |
| 4 | $\phi_4(\zeta) = \zeta^7 - 6.0363\zeta^6 + 15.4515\zeta^5 - 17.7016\zeta^4 + 8.8724\zeta^3 - 1.5534\zeta^2$ |

Free vibration analysis using these mode functions for a cantilever beam was performed.

The numerical results was compared with those of the analytical solution of Euler's beam [21] and are shown in Table 3. For reference, the comparison result in Table 3 is the calculated value of the relationship between the natural frequency of the beam and the properties.

**Table 3.** Comparison of FEM result and using a fourth-order polynomial.

|  | Fundamental $(\beta_n l)^2$ | Second $(\beta_n l)^2$ | Third $(\beta_n l)^2$ | Remark |
|---|---|---|---|---|
| FEM result | 3.51 | 21.99 | 61.44 | $(\beta_n l)^2 = w_n / \sqrt{\frac{EI}{\rho l^4}}$ |
| Using fourth-order polynomial | 3.51 | 22.00 | 61.70 | |

However, free vibration analysis for the L-type structure with a free end using these mode functions was not satisfactory, as shown in Figures 7 and 8.

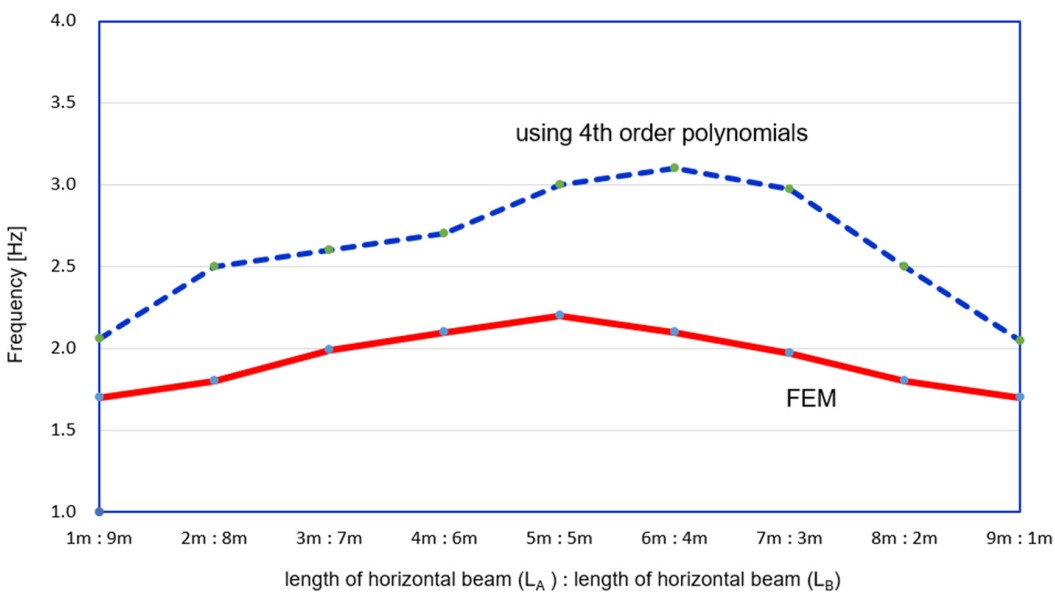

**Figure 7.** The comparison result of FEM and an L-type connected beam (using a fourth-order polynomial): First mode.

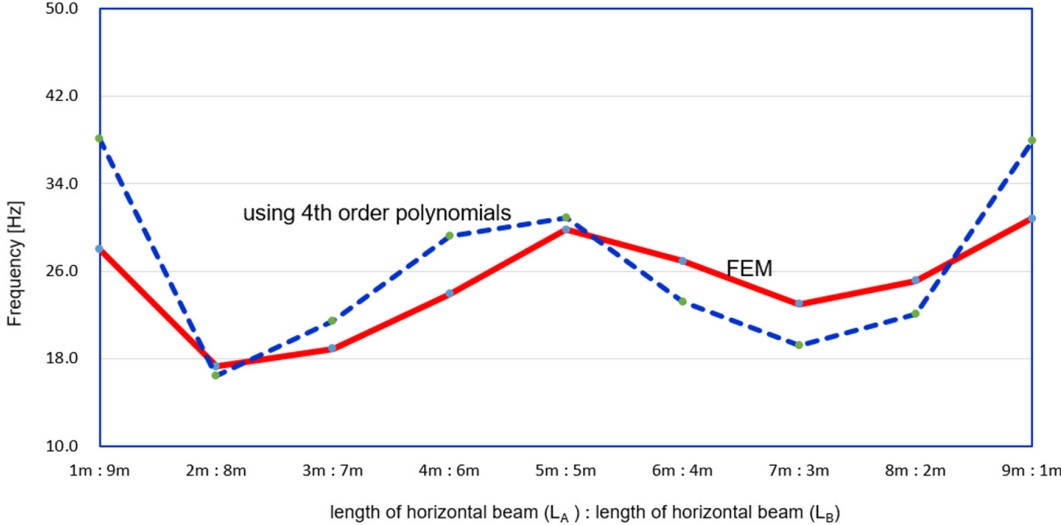

**Figure 8.** The comparison result of FEM and an L-type connected beam (using a fourth-order polynomial): Third mode.

## 5. Fundamental Mode Function Using Second-Order Polynomial (Adaptive Mode Function)

In order to make mode functions as flexible as possible, a second-order polynomial was chosen as a fundamental mode function, and higher orthogonal polynomials were generated, as suggested before by Bhat. In order to consider the rigid rotation of a vertical beam, $\psi_1(\xi) = \xi$ is added.

The lower four mode functions are shown Table 4.

**Table 4.** The mode function using adaptive polynomials.

| *i* and *j* | Horizontal Component ($\phi_i$) | Vertical Component ($\psi_j$) |
|:---:|:---:|:---:|
| 1 | $\zeta^2$ | $\xi$, |
| 2 | $\zeta^3 - 0.8333\zeta^2$ | $\xi^2$ |
| 3 | $\zeta^4 - 1.5\zeta^3 + 0.5357\zeta^2$ | $\xi^3 - 0.8333\xi^2$ |
| 4 | $\zeta^5 - 2.1\zeta^4 + 1.4\zeta^3 - 0.2917\zeta^2$ | $\xi^4 - 1.5\xi^3 + 0.5357\xi^2$ |

The numerical results are shown in Table 5, Table 6, Table 7. Thirteen mode functions, seven for $\varnothing_i$ and five for $\psi_j$, were used for this numerical analysis.

**Table 5.** Comparison of FEM results and using adaptive polynomials: First mode.

| Length ($L_A$:$L_B$) | 0:10 | 1:9 | 2:8 | 3:7 | 4:6 | 5:5 | 6:4 | 7:3 | 8:2 | 9:1 | 10:0 |
|:---:|:---:|:---:|:---:|:---:|:---:|:---:|:---:|:---:|:---:|:---:|:---:|
| FEM | 1.67 | 1.70 | 1.80 | 1.99 | 2.10 | 2.20 | 2.10 | 1.97 | 1.80 | 1.70 | 1.67 |
| Adaptive polynomial | 1.67 | 1.71 | 1.82 | 2.00 | 2.10 | 2.20 | 2.10 | 1.97 | 1.80 | 1.71 | 1.67 |

**Table 6.** Comparison of FEM results and using adaptive polynomials: second mode.

| Length ($L_A$:$L_B$) | 0:10 | 1:9 | 2:8 | 3:7 | 4:6 | 5:5 | 6:4 | 7:3 | 8:2 | 9:1 | 10:0 |
|:---:|:---:|:---:|:---:|:---:|:---:|:---:|:---:|:---:|:---:|:---:|:---:|
| FEM | 10.45 | 11.10 | 10.70 | 7.80 | 6.30 | 6.10 | 7.00 | 9.10 | 11.30 | 11.10 | 10.45 |
| Adaptive polynomial | 10.45 | 11.10 | 10.75 | 7.87 | 6.30 | 6.10 | 6.96 | 9.10 | 11.30 | 11.10 | 10.45 |

**Table 7.** Comparison of FEM results and using adaptive polynomials: third mode.

| Length ($L_A$:$L_B$) | 0:10 | 1:9 | 2:8 | 3:7 | 4:6 | 5:5 | 6:4 | 7:3 | 8:2 | 9:1 | 10:0 |
|:---:|:---:|:---:|:---:|:---:|:---:|:---:|:---:|:---:|:---:|:---:|:---:|
| FEM | 29.20 | 28.00 | 17.30 | 18.90 | 23.90 | 29.80 | 26.90 | 23.00 | 25.10 | 30.80 | 29.20 |
| Adaptive polynomial | 29.20 | 28.90 | 17.40 | 19.00 | 24.12 | 29.80 | 27.00 | 23.20 | 25.40 | 31.50 | 29.20 |

Typical corresponding natural modes are shown in Figure 9.

The numerical results showed very good agreement with the FEM results. This good agreement is due to the fact that suggested mode functions are flexible enough to follow anticipated deflections of an L-type beam with a free end. In that sense, we named these mode polynomials as having "adaptive mode function".

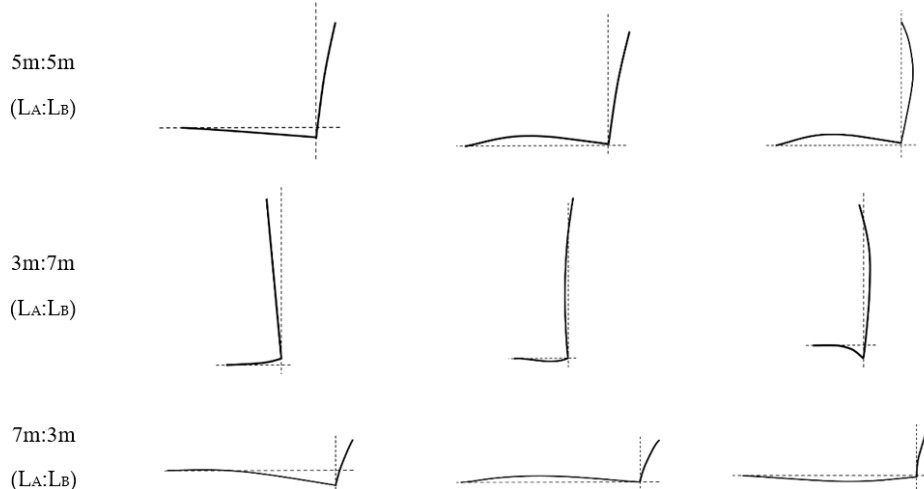

**Figure 9.** The mode shape of using adaptive polynomials.

## 6. Pure Mathematical Method

Mode functions for typical structures have been proposed, including this study [22,23]. There may be some structures where suitable mode functions may not be easy to generate.

In this case, a purely mathematical approach is suggested, where no meaningful higher-order mode functions are necessary. We may take mode functions in the following form:

$$W_A(\zeta) = \sum_{i=1}^{m} \phi_i(\zeta) p_i = \sum_{i=1}^{m} \zeta^{i+1} p_i \qquad (19)$$

$$W_B(\xi) = \sum_{j=1}^{n} \psi_j(\xi) q_j = \sum_{j=1}^{n} \xi^j q_j \qquad (20)$$

$$U_B(\xi) = r_1 \qquad (21)$$

Note that no higher-order mode functions are assumed. Most accurate natural frequencies are obtained using the how approach, although the eigenvectors obtained have no physical meaning.

This is due to the fact that no assumption for higher mode functions has been made. The numerical results were compared with those obtained from FEM analysis. Figure 1 was used for the calculation model and the beam properties mentioned in Table 1. The calculation results are shown in Table 8, Table 9, Table 10.

**Table 8.** Comparison of FEM results and using mathematical function: First mode.

| Length ($L_A$:$L_B$) | 0:10 | 1:9 | 2:8 | 3:7 | 4:6 | 5:5 | 6:4 | 7:3 | 8:2 | 9:1 | 10:0 |
|---|---|---|---|---|---|---|---|---|---|---|---|
| FEM | 1.67 | 1.70 | 1.80 | 1.99 | 2.10 | 2.20 | 2.10 | 1.97 | 1.80 | 1.70 | 1.67 |
| Mathematical function | 1.67 | 1.71 | 1.82 | 1.99 | 2.10 | 2.20 | 2.10 | 1.97 | 1.82 | 1.71 | 1.67 |

**Table 9.** Comparison of FEM results and using mathematical function: Second mode.

| Length ($L_A$:$L_B$) | 0:10 | 1:9 | 2:8 | 3:7 | 4:6 | 5:5 | 6:4 | 7:3 | 8:2 | 9:1 | 10:0 |
|---|---|---|---|---|---|---|---|---|---|---|---|
| FEM | 10.45 | 11.10 | 10.70 | 7.80 | 6.30 | 6.10 | 7.00 | 9.10 | 11.30 | 11.10 | 10.45 |
| Mathematical function | 10.45 | 11.16 | 10.75 | 7.87 | 6.34 | 6.08 | 6.96 | 9.10 | 11.33 | 11.15 | 10.45 |

**Table 10.** Comparison of FEM results and using mathematical function: Third mode.

| Length ($L_A$:$L_B$) | 0:10 | 1:9 | 2:8 | 3:7 | 4:6 | 5:5 | 6:4 | 7:3 | 8:2 | 9:1 | 10:0 |
|---|---|---|---|---|---|---|---|---|---|---|---|
| FEM | 29.20 | 28.00 | 17.30 | 18.90 | 23.90 | 29.80 | 26.90 | 23.00 | 25.10 | 30.80 | 29.20 |
| Mathematical function | 29.20 | 28.40 | 17.48 | 19.11 | 24.12 | 30.00 | 27.19 | 23.31 | 25.47 | 31.50 | 29.20 |

To better understand how good results can be obtained, use mode shapes together with eigenvectors. The mode shape is shown in Figure 10, while the eigenvectors are shown in Table 11.

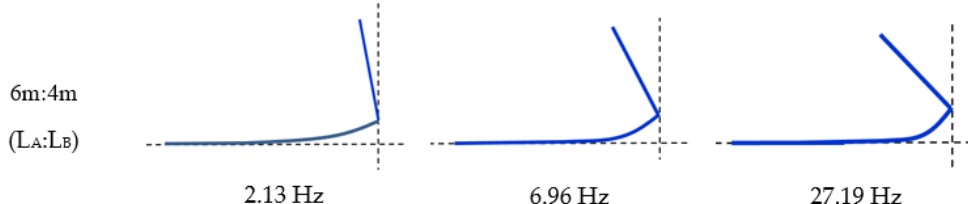

6m:4m

($L_A$:$L_B$)

2.13 Hz        6.96 Hz        27.19 Hz

**Figure 10.** Mode shapes using mathematical function.

**Table 11.** Eigenvectors of the mode shape.

| Coordinate | 2.1 Hz | 6.96 Hz | 27.2 Hz | Coordinate | 2.1 Hz | 6.96 Hz | 27.2 Hz |
|---|---|---|---|---|---|---|---|
| P1 ($\zeta^2$) | 1.00 | 1.00 | 1.00 | P11 ($\zeta^{12}$) | 2.23 | 26.60 | −15.65 |
| P2 ($\zeta^3$) | 1.10 | 3.78 | 2.65 | P12 ($\zeta^{13}$) | 2.36 | 29.07 | −18.33 |
| P3 ($\zeta^4$) | 1.22 | 6.45 | 2.36 | P13 ($\zeta^{14}$) | 2.49 | 31.54 | −21.04 |
| P4 ($\zeta^5$) | 1.34 | 9.05 | 1.08 | P14 ($\zeta^{15}$) | 2.62 | 34.00 | −23.75 |
| P5 ($\zeta^6$) | 1.46 | 11.61 | −0.75 | P15 ($\zeta^{16}$) | 2.76 | 36.47 | −26.48 |
| P6 ($\zeta^7$) | 1.58 | 14.14 | −2.91 | P16 ($\zeta^{17}$) | 2.89 | 38.93 | −29.23 |
| P7 ($\zeta^8$) | 1.71 | 16.65 | −5.28 | P17 ($\zeta^{18}$) | 3.02 | 41.39 | −31.99 |
| P8 ($\zeta^9$) | 1.84 | 19.15 | −7.78 | P18 ($\zeta^{19}$) | 3.16 | 43.84 | −34.77 |
| P9 ($\zeta^{10}$) | 1.97 | 21.64 | −10.36 | P19 ($\zeta^{20}$) | 3.29 | 46.30 | −37.57 |
| P10 ($\zeta^{11}$) | 2.10 | 24.12 | −12.99 | P20 ($\zeta^{21}$) | 3.42 | 48.75 | −40.39 |

## 7. Discussion

Our work deals with a very classic subject, and little research based on the assumed mode method has been found in last 20 years. Furthermore, free vibration analysis of an L-type beam with a free end is a typical example, even in the textbooks, for explaining component mode synthesis.

However, we believe that our work can renew appreciation of the usefulness of component mode method for free vibration analysis, by providing powerful mode functions.

As you can see. it will not be an easy task to find mode functions that can work on various configurations of L-type beam structures (length ratio $L_B/L_A$ varies from 0 to ∞). Certain mode functions that can work for one specific value of $L_B/L_A$ may not work for different values of $L_B/L_A$.

Although we do not include it in the paper, the suggested mode function comes from dozens of candidates. If a component is divided into subcomponents which may have geometrical boundary conditions only at one end, like a cantilever, then free vibration solutions may be very sensitive to the choice of mode functions. Most methods based on the Rayleigh–Ritz method use assumed modes.

However, we suggest using pure mathematical functions instead of using an assumed mode function. Although mode shapes using mathematical functions have nothing to do with real mode shapes, the results of the proposed method are compared with the FEM results and shown in Table 5, Table 6, Table 7, Table 8, Table 9, Table 10.

As a result, the function is accurate enough to show an error rate of less than 2% in all sections, regardless of the length ratio of the connected structure.

## 8. Conclusions

A second-order fundamental polynomial, together with higher orthogonal polynomials, is suggested as the most suitable assumed mode functions for an L-type beam structure with a free end.

The robustness of the suggested polynomials is proven through numerical analysis for an L-type beam structure with a free end against varying length ratios.

A purely numerical approach has been suggested for the structures where substructures have geometrical conditions only at one end, like a cantilever beam.

The most accurate natural frequencies are obtained this way, since any assumptions for higher-mode functions are unnecessary. Once natural frequencies are obtained, the way to find corresponding natural modes is worth studing.

**Author Contributions:** Conceptualization, D.Y.Y., J.H.P.; methodology, D.Y.Y., J.H.P.; software, J.H.P.; validation, J.H.P., D.Y.Y.; formal analysis, J.H.P., D.Y.Y.; investigation, J.H.P.; writing—original draft preparation, J.H.P., D.Y.Y.; writing—review and editing, J.H.P., D.Y.Y.; funding acquisition. All authors have read and agreed to the published version of the manuscript.

**Funding:** This research has been supported by the Chosun University research fund.

**Conflicts of Interest:** The authors declare no conflict of interest.

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
