# Peer review of "Adaptive Polynomials for the Vibration Analysis of an L-Type Beam Structure with a Free End"

_jmse, doi:10.3390/jmse9030300_

Round 1
Reviewer 1 Report
Suggestion of adaptive polynomials for the vibration 2 analysis of L-type beam structure with a free end
By Duck Young Yoon, Jeong Hee Park
The manuscript needs to improve wording and grammar standard, such as:
P1
2 Suggestion of adaptive…
Adaptive…
26 Two component mode method are well known
Two component mode methods are well known
69 simplify of one end supported structure -- grammar incorrect.
Etc.
The structure of paper can be improved to present the accuracy of modal synthesis to show ‘2nd order fundamental polynomial together with higher orthogonal polynomials are suggested as the most suitable assumed mode functions for L-type beam structure with a free end’
Author Response
2 Suggestion of adaptive…
Reply) The title has been modified as suggested.
26 Two component mode method are well known
Reply) Corrected the word as indicated.
69 simplify of one end supported structure -- grammar incorrect.
Reply) Corrected the pointed out.
The structure of paper can be improved to present the accuracy of modal synthesis to show ‘2nd order fundamental polynomial together with higher orthogonal polynomials are suggested as the most suitable assumed mode functions for L-type beam structure with a free end’
Reply) Lowest 4 mode functions suggested are shown on table 4.
Reviewer 2 Report
Thanks for your contribution to the JMSE journal.
This manuscript suggests adaptive polynomials for the vibration analysis of L-type beam analysis.
Basically, it is well described, but some part can be improved based on review comments.
[Comment 1]
[Title, Discussion & Conclusion] Title is a summary of the abstract. In conclusion, this the suggested idea and its effect shall be highlighted based on the obtained validation outcome. Perhaps, statistical analysis or other comparison results can be added in the discussion part to support author's suggestion.
[Comment 2] Introduction
It is too brief to introduce current issues on this topic (including historical and technical reviews). Authors could add additional state-of-the-art research outcomes.
[Comment 3] Section 3 FEM analysis
For the FE analysis, more detailed design input, FE technique, element type, geometric and material properties, etc. should be provided.
[Comment 4] Analysis results are well presented. However, the accuracy can be explained by statistical analysis results, i.e., R2 or Mean or COV or others.
[Comment 5] Before the conclusion, the discussion part can be added to highlight the strength of the author's suggestion.
[Minor Comments] English language and style are fine and minor spell check required. Journal format and references should also be checked.
Author Response
[Comment 1]
[Title, Discussion & Conclusion] Title is a summary of the abstract. In conclusion, this the suggested idea and its effect shall be highlighted based on the obtained validation outcome. Perhaps, statistical analysis or other comparison results can be added in the discussion part to support author's suggestion.
[Reply] As pointed out, a discussion section was added before the conclusion to emphasize the methodology based on the results obtained for the method proposed in this study.
[Comment 2] Introduction
It is too brief to introduce current issues on this topic (including historical and technical reviews). Authors could add additional state-of-the-art research outcomes.
[Reply] In the introduction section, the background of the research and the results of recent research have been added.
[Comment 3] Section 3 FEM analysis
For the FE analysis, more detailed design input, FE technique, element type, geometric and material properties, etc. should be provided.
[Reply] The model and geometric conditions used in the FEM analysis are displayed in section 3.
[Comment 4] Analysis results are well presented. However, the accuracy can be explained by statistical analysis results, i.e., R2 or Mean or COV or others.
[Reply] The calculation results are presented in tables.
[Comment 5] Before the conclusion, the discussion part can be added to highlight the strength of the author's suggestion.
[reply] As pointed out, the discussion section was added to emphasize the part suggested in the paper.
[Minor Comments] English language and style are fine and minor spell check required. Journal format and references should also be checked.
[Reply] We checked and corrected the English spelling for the pointed out, and we checked the sources again for the reference literature.
Reviewer 3 Report
The authors are working on very classic subject in mechanics of materials, which has important practical meaning to structural designers. From the references, it is seen that most of them are very old, with a few exceptions dated after 2000. It can be two reasons, either the subject is nowadays very rare or the authors are not aware of the latest development. As such, it is difficult for the authors to define the originality and novelty of their work.
Overall, the presented work is some comparison between classic simplified methods and numerical results out of finite element analysis. In my opinion, it is some meaningful material for discussion at a conference, but not at the level of a journal paper publication.
The legends in several graphs are not shown correctly, as in the graphs it is always a combination of lines and markers.
There is also a typo in Figure 6, FEA should be FEM.
Author Response
The authors are working on very classic subject in mechanics of materials, which has important practical meaning to structural designers. From the references, it is seen that most of them are very old, with a few exceptions dated after 2000. It can be two reasons, either the subject is nowadays very rare or the authors are not aware of the latest development. As such, it is difficult for the authors to define the originality and novelty of their work.
[Reply] As you commented our work deals with very much classic subject and little researches based on assumed mode method have been found in last 20 years. Furthermore free vibration analysis of L type beam with a free end a typical example even in the text books for explaining component mode synthesis.
However, we believe that our work can renew appreciation of usefulness of component mode method for free vibration analysis by provide powerful mode functions.
Since you agree the practical meaning of our work, we reply what originality and novelty our work has.
1) As you can see it will not be easy a task to find mode functions which can work on various configuration of L type beam structure (Length ratio varies from 0 to ).
Certain mode functions which can work for one specific values of may not work for different values of
Although we do not include in the paper, the suggested mode functions comes from dozens of candidates.
2) If component is divided into subcomponents which may have geometrical boundary conditions only at one end like cantilever, free vibration solutions may be very much sensitive to the choice of mode functions. Most of method based on Rayleigh-Ritz method use assumed modes.
However, we suggest to use pure mathematical functions instead of using assumed mode function.
Although mode shapes using mathematical functions have nothing to do with real mode shapes, exact natural frequencies can be found.
Overall, the presented work is some comparison between classic simplified methods and numerical results out of finite element analysis. In my opinion, it is some meaningful material for discussion at a conference, but not at the level of a journal paper publication.
[Reply] We appreciate it if you realize the meaning of our research again through our reply.
The legends in several graphs are not shown correctly, as in the graphs it is always a combination of lines and markers.
Reply) Taking into account your comment, I replaced the graph with a table.
There is also a typo in Figure 6, FEA should be FEM.
Reply) Corrected the pointed out.
Round 2
Reviewer 3 Report
The introduction part can be strengthened to describe the stare-of-art of the research on the subject.
Author Response
The introduction part can be strengthened to describe the stare-of-art of the research on the subject.
Reply) As you commented, several references are added, which mainly deal with nonlinear mechanics.